# EXAMINING SCALING AND TRANSFER OF LANGUAGE MODEL ARCHITECTURES FOR MACHINE TRANSLATION

## ABSTRACT

Natural language understanding and generation models follow one of the two dominant architectural paradigms: *language models* (LMs) that process concatenated sequences in a single stack of layers, and *encoder-decoder models* (EncDec) that utilize separate layer stacks for input and output processing. In machine translation, EncDec has long been the favoured approach, but with few studies investigating the performance of LMs. In this work, we thoroughly examine the role of several architectural design choices on the performance of LMs on bilingual, (massively) multilingual and zero-shot translation tasks, under systematic variations of data conditions and model sizes. Our results show that: (i) Different LMs have different scaling properties, where architectural differences often have a significant impact on model performance at small scales, but the performance gap narrows as the number of parameters increases, (ii) Several design choices, including causal masking and language-modeling objectives for the source sequence, have detrimental effects on translation quality, and (iii) When paired with full-visible masking for source sequences, LMs could perform on par with EncDec on supervised bilingual and multilingual translation tasks, but improve greatly on zero-shot directions by facilitating the reduction of off-target translations.

## 1 INTRODUCTION

The popularity of large, general-purpose text generation models has skyrocketed in recent years due to their outstanding performance across a wide range of natural language processing tasks (Brown et al., 2020; Raffel et al., 2020; Liu et al., 2020; Xue et al., 2021). These studies, under large-scale pretraining and also model scaling, show the promise of moving to unified neural architectures and that dropping various task-specific inductive biases improves model generalization and/or performance (Devlin et al., 2019; Kaplan et al., 2020). In neural machine translation (NMT), one essential task-specific inductive bias is to separately handle source sentence understanding and target sentence generation with the encoder-decoder paradigm (EncDec). Although such bias significantly benefits translation (Vaswani et al., 2017; Chen et al., 2018; Aharoni et al., 2019; Barrault et al., 2020; Ansari et al., 2020), some recent work shows insights challenging it, for example, aggressively simplifying the decoder yields little to no compromise on translation quality (Kasai et al., 2021). This thereby inspires the question how the removal of this separation, i.e. using a single unified module for both encoding and decoding (LMs), works for translation, and whether we can get any benefits out of that.

Although some studies reported promising translation quality with LMs (He et al., 2018; Wang et al., 2021), they compare models under merely one configuration (model size), neglecting that neural models follow scaling laws (Kaplan et al., 2020; Ghorbani et al., 2021; Gordon et al., 2021) where the impact of each added parameter on model performance might vary across different models. How the inductive biases of LMs and EncDec impact the model's performance as we increase their size and the amounts of training data are intriguing yet missing in the literature.

In this paper, we explore various configurations of LM architectures for translation as in Figure 1, and compare them with EncDec on model scaling and cross-lingual transfer. We conduct a systematic study under a variety of data conditions, tasks (bilingual, multilingual and zero-shot) and model

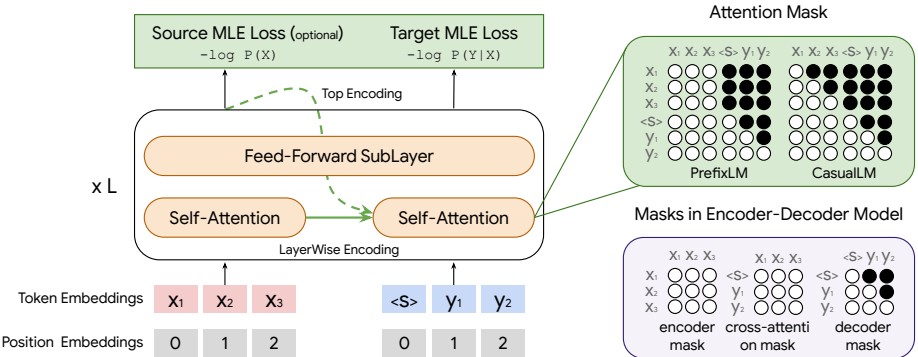

Figure 1: Illustration for translation-oriented language models. $X$ and $Y$ denote source and target input, respectively. To enable translation, we adapt the LM self-attention mask to either the PrefixLM mask or CausalLM mask (top right), where filled black circles indicate disallowed attention. We also explore top-only encoding (Top Encoding) for PrefixLM which feeds the final-layer source encodings to generation similar to EncDec, rather than layer-wise coordinated encodings (He et al., 2018). Masks of EncDec are shown in the bottom right for comparison.

capacities. We examine architectural design choices associated with LMs, including causal masking (CausalLM) vs. full-visible masking (PrefixLM) for source sequences,[1] layer-wise coordination (He et al., 2018) vs. final-layer source encodings (TopOnly) for target sequence generation, increasing LM depth vs. width, and also the effect of adding source language modeling loss for CausalLM.

We evaluate how these architectural priors affect translation quality as we increase the number of parameters available to the model, under a diverse set of bilingual translation settings and also (massively) multilingual settings, with a special focus on transfer to low-resource languages and zero-shot directions. Our main findings are summarized below:

- LMs show different scaling properties compared to EncDec. The architectural differences become less important as models scale, measured by reduced quality gap against EncDec, regardless of the language similarities, training data conditions and evaluation settings.

- PrefixLM variants often outperform their CausalLM counterparts; increasing LM depth benefits the translation task more than increasing the width; and adding a source-side language modeling objective to CausalLM doesn't affect translation quality.

- Cross-lingual transfer also benefits from model scaling, where EncDec almost always dominates the quality Pareto frontier on supervised directions while zero-shot translation favors PrefixLM LMs. PrefixLM LMs significantly reduce off-target translations.

- Although LMs could reach or even surpass the translation quality of EncDec, they still lag far behind EncDec with respect to computational efficiency as measured in FLOPs.

## 2   RELATED WORK

Using language models in the task of translation has a long history, particularly in the era of statistical machine translation (SMT) where LM was used as a separate yet crucial component ensuring the fluency of generation (Stolcke, 2002; Heafield, 2011; Koehn, 2010). With neural networks, NMT unified those isolated SMT components including LM under the encoder-decoder formulation (Kalchbrenner & Blunsom, 2013; Cho et al., 2014; Sutskever et al., 2014; Bahdanau et al., 2015), which makes use of separate modules to process input and output. Further studies exploring architectural modifications by using LM alone as a translation model, nevertheless, got much less attention. He et al. (2018) proposed *layer-wise coordination* between encoder and decoder with tied weights, where each decoder layer attends to its corresponding encoder layer at the same depth as opposed to the conventional method of attending the top-most encoder representations. Later,

---

[1]Also known as unidirectional vs bidirectional language modelling, where in the unidirectional case a token representation takes into account only the preceding tokens and their representations, but the bidirectional case takes into account both preceding and following tokens in a sequence.

| Model | Objective | | Structure | | Src-Src | Parameter |
| --- | --- | --- | --- | --- | --- | --- |
| | $-\log P(X)$ | $-\log P(Y\mid X)$ | Layer-Wise | TopOnly | Mask | Sharing |
| EncDec | | ✓ | | ✓ | Full | ✗ |
| PrefixLM | | ✓ | ✓ | | Full | ✓ |
| + TopOnly | | ✓ | | ✓ | Full | ✓ |
| CausalLM | ✓ | ✓ | ✓ | | Causal | ✓ |
| + TgtOnly | | ✓ | ✓ | | Causal | ✓ |

Table 1: Comparison of different models. $X/Y$: source/target input. *Layer-Wise*: layer-wise coordination; *TopOnly*: use topmost-layer source encodings; *Src-Src Mask*: the intra-source masking schema, either fully visible (Full) or causal (Causal); *Parameter Sharing*: share parameters between source and target.

Fonollosa et al. (2019) extended it with locality constraint. Dong et al. (2019) explored LMs for sequence generation under large-scale pretraining. Despite reporting promising results, these prior studies either focus only on bilingual tasks or do not consider the scaling properties of the models, leaving the picture incomplete: how the findings will change as we scale the models and how the languages benefit from/interfere each other as the architectural priors (inductive biases) change.

Neural models follow some scaling laws. Kaplan et al. (2020) reported the test cross-entropy loss of LMs can be formulated as a power-law scaling function of either model size (excluding embedding parameters) or dataset size. Later on, researchers examined and confirmed such findings across different domains, including vision modeling (Zhai et al., 2021), knowledge transfer from pretraining (Hernandez et al., 2021), autoregressive generative modeling (Henighan et al., 2020), and neural machine translation (Gordon et al., 2021; Ghorbani et al., 2021), to name a few. We find it essential to study the scaling behavior of new architectures and approaches given the recent evidence on the emergent properties of the models at scale (Brown et al., 2020).

Another critical component in machine translation is the number of languages being considered with the models, which is the very focus of multilingual NMT (Firat et al., 2016). Cross-lingual transfer in multilingual NMT often results from parameter sharing across languages, which benefits low-resource languages and also enables zero-shot translation (Johnson et al., 2017), although the quality on zero-shot directions is largely hindered by the off-target translation problem (Arivazhagan et al., 2019; Zhang et al., 2020). The structure of LMs further encourages parameter sharing, offering a chance to improve the transfer while magnifying the problem of interference (negative-transfer) (Wang et al., 2020; Zhang et al., 2021). Very recently, Wang et al. (2021) analyzed the cross-lingual transfer behavior of CausalLM, and reported encouraging zero-shot performance. However, we did not observe the same results likely because of data sampling, model architecture and optimization differences which zero-shot transfer is sensitive to.

## 3 LANGUAGE MODEL ARCHITECTURES FOR TRANSLATION

In this section, we briefly review EncDec and then present LM architectures for translation based on Transformer (Vaswani et al., 2017). Table 1 compares different models. Given a source sequence $\mathbf{X}$ of length $|X|$ and its target translation $\mathbf{Y}$ of length $|Y|$, EncDec performs translation via the following structure:

$$\mathbf{X}^l = \text{FFN} \circ \text{SAtt}\left(\mathbf{X}^{l-1}\right), \quad \mathbf{Y}^l = \text{FFN} \circ \text{CAtt} \circ \text{SAtt}\left(\mathbf{Y}^{l-1}, \mathbf{X}^L\right), \quad (1)$$

where $l$ denotes the layer index and $\circ$ indicates consecutive sublayers. $\mathbf{X}^l \in \mathbb{R}^{|X| \times d}$ and $\mathbf{Y}^l \in \mathbb{R}^{|Y| \times d}$ are the layer representations of the source and target sequence respectively, with a model dimension of $d$. The first input layer $(\mathbf{X}^0, \mathbf{Y}^0)$ is the summation of token embeddings and their positional encodings. We drop all the layer normalization and residual connections in our formulations for brevity.

The encoder is a stack of $L$ layers, each of which includes a multi-head self-attention sublayer (SAtt) followed by a feed-forward sublayer (FFN). SAtt in the encoder is bidirectional with *full-visible masking* that has full visibility to all source tokens, preceding and following. Its final-layer representations $\mathbf{X}^L$ are fed to the decoder, which shares a similar structure to the encoder but with

| Dataset | #Samples (Sources) | | | Experiments | |
|---|---|---|---|---|---|
| | Train | Dev | Test | BIL | MUL |
| WMT14 En-De | 4.5M | 3000 (WMT13) | 3003 (WMT14) | | ✓ |
| WMT14 En-Fr | 41M | 3000 (WMT13) | 3003 (WMT14) | ✓ | ✓ |
| WMT19 En-Zh | 26M | 3981 (WMT18) | 1997 (WMT19, SO) 2000 (WMT19 TO) | ✓ | ✓ |
| Web En-De | 2B | 7927 (Web) | 4927/1997 (Web/WMT19, SO) 6000/2000 (Web/WMT19, TO) | ✓ | |

Table 2: Statistics of different datasets. *M/B*: million/billion; *SO/TO*: source-original/target-original test sets; *Web*: in-house web-crawled datasets; *BIL/MUL*: the data is used for bilingual/multilingual experiments.

an additional (multi-head) cross-attention sublayer (CAtt). Unlike encoder, SAtt in the decoder is unidirectional with *causal masking*, where attention to following tokens is disabled (masked). CAtt can always access all source inputs, though. Note we set the encoder and decoder depth equally to $L$, and use $d^{\mathrm{ff}}$ to denote the intermediate dimension of FFN. EncDec is often optimized with the target translation objective based on $\mathbf{Y}^L$:

$$\mathcal{L}^{\mathrm{EncDec}}(X, Y) = \mathcal{L}^{\mathrm{TGT}} = -\log P(Y|X, \mathbf{Y}^L). \tag{2}$$

Rather than separately modeling source and target sequences, LM handles both with a single module:

$$\left[\mathbf{X}^l, \mathbf{Y}^l\right] = \mathrm{FFN} \circ \mathrm{SAtt}\left(\left[\mathbf{X}^{l-1}, \mathbf{Y}^{l-1}\right], \mathbf{M}\right), \tag{3}$$

where $\mathbf{M} \in \{0, 1\}^{(|X|+|Y|)\times(|X|+|Y|)}$ is the attention mask that controls the information flow within the concatenated sequences ($[\cdot, \cdot]$).[2] Two LM variants explored by changing the structure of mask $\mathbf{M}$, *PrefixLM* and *CausalLM*.

**PrefixLM** merges different modules of EncDec, trained with $\mathcal{L}^{\mathrm{TGT}}$. Its attention mask

$$\mathbf{M}^{\mathrm{PrefixLM}}(i, j) = 1, \text{ if } i \geq j \text{ or } j \leq |X|; \text{ otherwise } 0, \tag{4}$$

combines the encoder/decoder self-attention mask and the cross-attention mask of EncDec.

**CausalLM**, by contrast, is a strict LM that applies causal masking to both sequences:

$$\mathbf{M}^{\mathrm{CausalLM}}(i, j) = 1, \text{ if } i \geq j; \text{ otherwise } 0. \tag{5}$$

Apart from $\mathcal{L}^{\mathrm{TGT}}$, CausalLM also includes the source-side language modeling loss for training:

$$\mathcal{L}^{\mathrm{CausalLM}}(X, Y) = \mathcal{L}^{\mathrm{SRC}} + \mathcal{L}^{\mathrm{TGT}} = -\log P(X|\mathbf{X}^L) - \log P(Y|X, \mathbf{Y}^L). \tag{6}$$

To improve our understanding of LMs for translation, we further incorporate two extensions:

**PrefixLM + TopOnly** The model defined in Equation 3 performs attention over the source and target sequence within the same layer. In contrast, EncDec always uses the topmost-layer source encodings for translation. We mimic this with *TopOnly* extension by feeding top-layer encodings, i.e. $\mathbf{X}^L$ instead of $\mathbf{X}^{l-1}$, to each attention sublayer. It operates the same as EncDec but with the parameters of encoder and decoder tied.

**CausalLM + TgtOnly** The inclusion of the source-side objective enriches CausalLM's learning signal and encourages the model to absorb source language characteristics. However, it requires and occupies part of modeling capacity, which might negatively affect translation. To offset this impact, we add the *TgtOnly* extension that optimizes CausalLM with the target translation objective $\mathcal{L}_C^{TGT}$ alone, which also aligns better with EncDec and PrefixLM.

## 4 SETUP

**Model Setting** We use Transformer for experiments. By default, we adopt the base setting, with $d = 512$, $d^{\mathrm{ff}} = 2048$ and 8 attention heads. We also work with the Transformer big setting where each hyper-parameter above is doubled. Training and inference details are in Appendix A.

---

[2]Note that, in our implementation we still use separate source and target positions as shown in Figure 1.

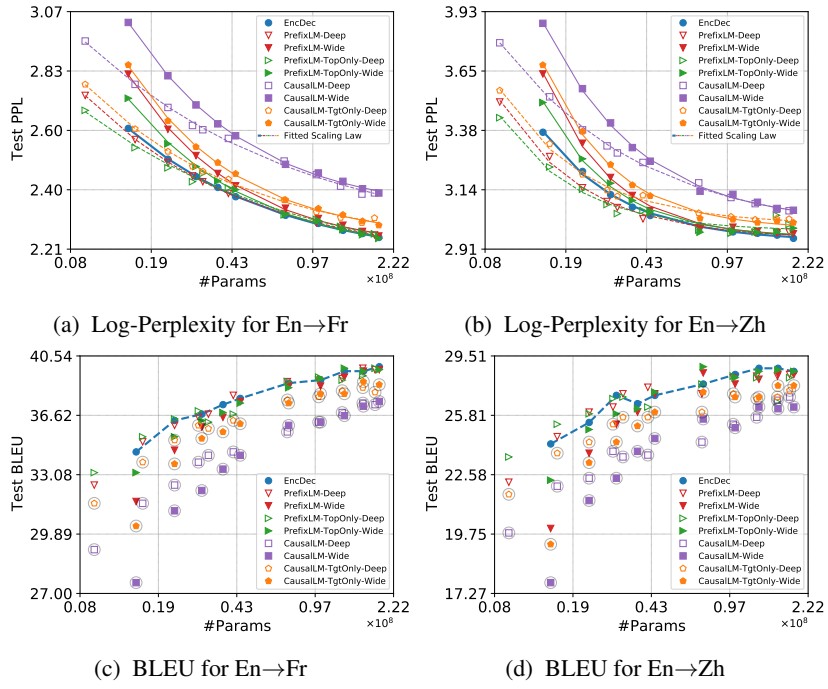

(a) Log-Perplexity for En→Fr

(b) Log-Perplexity for En→Zh

(c) BLEU for En→Fr

(d) BLEU for En→Zh

Figure 2: Fitted scaling curves (top) and BLEU scores (bottom) for different models on WMT14 En-Fr (left) and WMT19 En-Zh (right) tasks. **Top**: dashed and solid fitted curves are for *LM + Deep* and *LM + Wide*, respectively. We represent the EncDec scaling with bold solid curve. **Bottom**: dashed curve denotes the BLEU scores of EncDec as a function of model parameters for reference. Markers in circles are for CausalLM variants. Models are trained in Transformer base setting. Best seen in color.

**Datasets and Evaluation** We use WMT14 English-French (En-Fr), WMT14 English-German (En-De), WMT19 English-Chinese (En-Zh) and an in-house web-crawled (Web) En-De dataset for experiments, whose statistics are summarized in Table 2. We also report results on OPUS-100 (Zhang et al., 2020), a massively multilingual corpus containing 100 languages. All datasets are pre-processed with byte pair encoding (Sennrich et al., 2016, BPE) implemented by Sentence-Piece (Kudo & Richardson, 2018). We set the BPE vocabulary size to 32K by default. We report test log-perplexity score (PPL) for scaling study particularly and also show SacreBLEU (Post, 2018)[3].

## 5 EXPERIMENTS FOR MODEL SCALING

Kaplan et al. (2020) reported the model performance can be described with a power-law, with respect to its parameters, as below:

$$\mathcal{L}(N) = \alpha \left( \frac{N_0}{N} \right)^p + \mathcal{L}_\infty, \tag{7}$$

where $\mathcal{L}(N)$ fits test PPL, and $N$ denotes the number of parameters. $N_0$ is a constant used for numerical stability which is obtained from 1-layer EncDec model. $\alpha, p, \mathcal{L}_\infty$ are fitted parameters, and we mainly analyze the estimated scaling exponent $p$ and the irreducible loss $\mathcal{L}_\infty$ for different models.

The way of increasing model parameters varies for the same model and also across different models. We perform scaling firstly for EncDec by changing the depth $L$ (from 1 to 26 layers, equally for its encoder and decoder) while keeping the other hyper-parameters intact following Ghorbani et al. (2021). We then align the scaling settings of LM with its EncDec counterpart in term of model parameters through increasing either its depth or width:

- *LM + Deep* adds parameters via stacking more Transformer layers, which was also used in previous studies (He et al., 2018; Wang et al., 2021).

---

[3]Signature: *BLEU+case.mixed+lang\*+numrefs.1+smooth.exp+tok.13a+v.1.5.1*

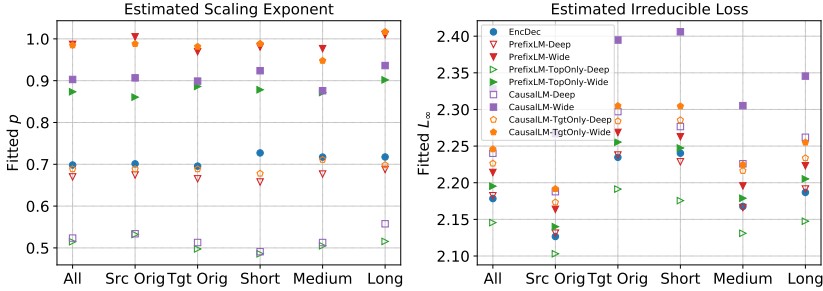

Figure 3: Fitted scaling exponent ($p$, left) and irreducible loss ($\mathcal{L}_\infty$, right) over different evaluation settings on WMT14 En-Fr (En→Fr). *All*: the whole test set; *Src Orig*, *Tgt Orig*: source-original and target-original test set, respectively; *Short, Medium, Long*: shortest, medium and longest ∼376 samples from the test set, respectively.

- *LM + Wide*, instead, grows the model width. We choose to enlarge the feed-forward dimension from $d^{\mathrm{ff}}$ to $3d^{\mathrm{ff}}$. Note other strategies for width scaling are possible and many, but exploring them is resource-consuming and beyond the scope of our paper.

We distinguish data-limited regime from model size-limited regime for model scaling (Bahri et al., 2021), where the former has relatively fewer training samples than model parameters thus likely suffers from overfitting (e.g. with WMT14 En-Fr and WMT19 En-Zh), while the latter has enough samples for model fitting (e.g. with Web En-De).

## 5.1 Scaling in Data-Limited Regime

**Architectural difference matters most when the model is at a small scale.** Figure 2 summarizes the scaling results on WMT14 En-Fr and WMT19 En-Zh. When there are fewer parameters, the model with inductive biases favoring translation will achieve better quality. Such inductive bias includes 1) allowing the full visibility to the source input as in PrefixLM[4] rather than causal masking; 2) using topmost-layer source encodings for translation (*TopOnly*) rather than layer-wise coordinated encodings; 3) deeper LMs (*Deep*) rather than wider; and 4) training LMs without source-side language modeling loss (*TgtOnly*). The fact that *LM + Deep* outperforms *LM + Wide* shows that not only the number of parameters matters, but also the way parameters are added. This aligns with the previous findings: deeper models apply more non-linear operations and induce more abstract representations, which often helps translation (Wang et al., 2019). This also applies to TopOnly. Most of these findings are consistent across different languages and evaluation metrics (Figure 2a-2d).

**Different models show different scaling properties, but the gap narrows at scale.** The impact of added parameters on translation performance differs across different models. The LMs performing poorly at small scale often gain more from the increased capacity. For instance, the difference between *LM + Deep* and *LM + Wide* almost disappears at the end, resonating with the optimal depth-vs.-width theory (Levine et al., 2020). We observe that PrefixLM and EncDec converge to a similar region, followed by *CausalLM + TgtOnly* while *CausalLM* still retains a clear gap against the others. This performance gap on WMT19 En-Zh is smaller, mainly because of model overfitting. BLEU scores in Figure 2c and 2d also show such a trend, although the relationship between BLEU and PPL is non-trivial (Ghorbani et al., 2021). These tell us that the success of architectural modifications on small-scale models might not transfer to large-scale settings, and that comparing different models under one model configuration might result in incomplete and misleading conclusions. Note we also observe reduced gap when considering the number of layers (see Figure 9 in the Appendix).

**Do sentence length and originality of test samples affect scaling properties?** Not much! We further test how the scaling changes across different evaluation settings, and show the results on WMT14 En-Fr in Figure 3. The scaling exponent changes marginally over different settings (often less than 0.05), suggesting that the scaling curves are quite similar in these settings (see Figure 8, 10, 11 in Appendix), although sentences of different originalities differ largely in style and natural-

---

[4]By default, we use PrefixLM (CausalLM) to refer to all PrefixLM variants (CausalLM variants). We adopt the *italic* form to denote a specific variant.

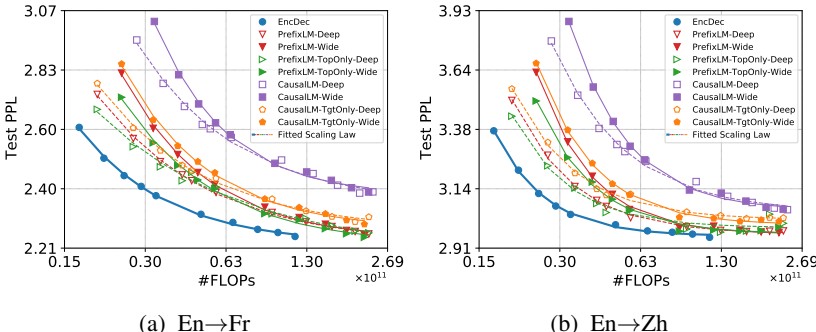

(a) En→Fr        (b) En→Zh

Figure 4: Fitted scaling curves for different models on WMT14 En-Fr and WMT19 En-Zh in term of FLOPs.

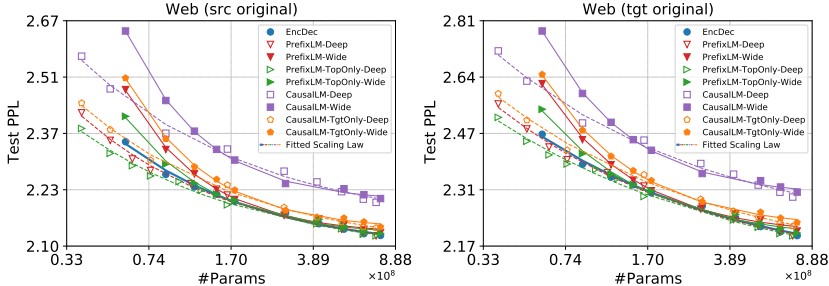

Figure 5: Fitted scaling curves for different models on Web En-De (En→De). *src/tgt*: source/target; *Web*: in-domain evaluation set. Models are trained in the Transformer big setting.

ness (Graham et al., 2020; Freitag et al., 2020). The estimated irreducible loss shows that target-original parallel sentences are harder to model than the source-original ones, and that translating medium-length sentences is much easier. The loss ranking of different models changes little over these settings, supporting PrefixLM and EncDec generally more than CausalLM.

**Do LMs deal with computational efficiency better than EncDec?** No! The FLOPs results in Figure 4 show that EncDec demands generally less computation than LM, but the gap narrows at scale. Note LM doesn't save computations. By contrast, to perform similarly to EncDec, LM often goes wider or deeper, which even deteriorates its running (training and decoding) efficiency. Besides, EncDec allows arbitrary decoders, e.g. shallow decoders for faster inference, which is non-feasible for LMs. Figure 4 also shows that adding the source-side loss hurts CausalLM's efficiency.

## 5.2 SCALING IN MODEL SIZE-LIMITED REGIME

Figure 5 shows the in-domain scaling performance on Web En-De. Overall, we observe similar scaling patterns as discovered above, and such pattern transfers to out-of-domain evaluation, FLOPs and BLEU scores. More results are available in the Appendix (Figure 12, 13 and 14).

## 6 EXPERIMENTS FOR CROSS-LINGUAL TRANSFER

Based on the literature (Wang et al., 2020; Zhang et al., 2021), sharing capacity across languages could encourage knowledge transfer but might also gain the risk of negative interference. In this section, we further compare different models but on multilingual many-to-many translation. To enable multilingual NMT, we append a target language tag to each source sentence following Johnson et al. (2017). We perform over-sampling to balance the training data with a temperature of $T = 5$.

**Do LMs facilitate the transfer to low-resource languages?** Not really! We start with multilingual translation for WMT En-De/Fr/Zh, and regard En-De as a relatively low-resource language pair. One reason behind the popularity of multilingual NMT is its transfer capability to low-resource languages. We analyze this transfer behavior for LMs and explore transfer (to De) from similar (Fr) and

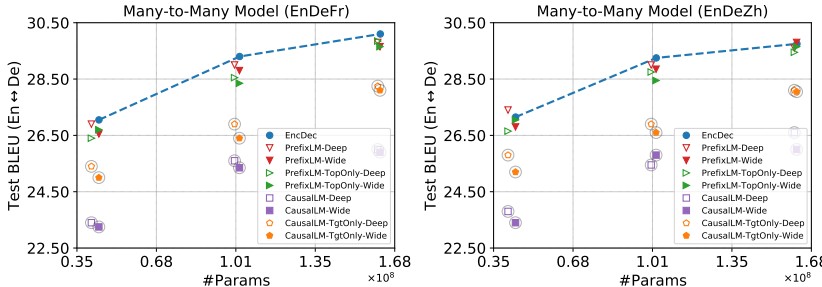

Figure 6: Cross-lingual transfer results (average BLEU scores) for different models from high-resource languages to the low-resource one (En-De) under different model sizes on WMT datasets. Average is performed over En→De and De→En evaluation. **Left**: multilingual En-De-Fr system; **Right**: multilingual En-De-Zh system. Both systems are many-to-many models. Models are trained in the Transformer base setting.

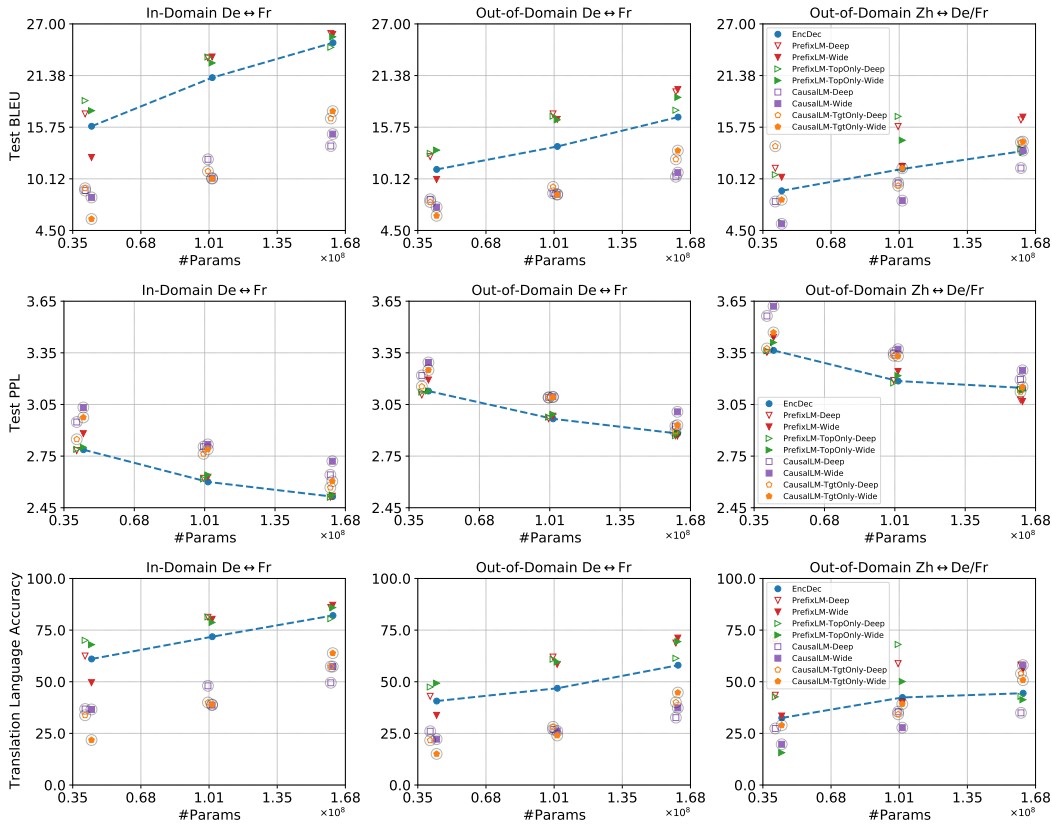

Figure 7: Zero-shot transfer results of different models for multilingual many-to-many modeling on four languages (En-De-Fr-Zh) under different model sizes. **Top**: average BLEU scores; **Middle**: average PPL scores; **Bottom**: average translation language accuracy scores. *In-domain*: WMT test set; *Out-of-domain*: in-house sport-domain test sets.

distant (Zh) languages separately. Figure 6 shows the results. PrefixLM produces comparable results to EncDec, while CausalLM lags far behind, and the incorporation of source-side objective actually hurts translation. Overall, we observe that EncDec almost dominates the transfer performance under different model sizes, regardless of language similarity. Similar results are also observed for low-resource to high-resource transfer (see Figure 15 in the Appendix).

**Do LMs benefit zero-shot transfer?** PrefixLM does! We further test how LMs perform on zero-shot translation. We use the newstest2019 De-Fr test set as the in-domain zero-shot eval set, and an internal sports-domain N-way test set for De-Fr-Zh (2000 samples) as the out-of-domain eval

| | Model | En→XX | | | | XX→En | | | | Zero-Shot | |
|---|---|---|---|---|---|---|---|---|---|---|---|
| | | High | Med | Low | All | High | Med | Low | All | BLEU | ACC |
| | EncDec | 25.8 | 32.4 | 31.9 | 29.2 | 31.4 | 34.3 | 35.0 | 33.1 | 4.80 | 24.21 |
| D | PrefixLM | -0.34 | -0.21 | -0.82 | -0.41 | -0.27 | -0.74 | -1.59 | -0.70 | 7.95 | 41.46 |
| | + TopOnly | -0.01 | -0.14 | -1.79 | -0.44 | -0.07 | -0.71 | -1.43 | -0.57 | 6.59 | 39.06 |
| | CausalLM | -4.51 | -8.18 | -12.9 | -7.47 | -5.18 | -10.1 | -13.0 | -8.38 | 4.10 | 25.60 |
| | + TgtOnly | -0.83 | -0.78 | -1.40 | -0.93 | -1.27 | -1.81 | -2.43 | -1.69 | 7.34 | 39.62 |
| W | PrefixLM | -0.71 | -0.75 | -2.02 | -1.01 | -0.77 | -0.88 | -0.68 | -0.78 | 7.44 | 38.60 |
| | + TopOnly | -0.40 | -0.37 | -0.66 | -0.45 | -0.47 | -0.50 | -1.41 | -0.69 | 6.92 | 37.69 |
| | CausalLM | -4.25 | -7.58 | -12.2 | -7.03 | -5.05 | -9.88 | -13.3 | -8.32 | 4.49 | 28.08 |
| | + TgtOnly | -1.29 | -1.27 | -0.82 | -1.18 | -1.88 | -1.96 | -2.04 | -1.94 | 5.53 | 29.75 |

Table 3: Translation quality of different models for En→XX, XX→En and zero-shot language pairs on OPUS-100. Models are trained in the Transformer big setting, aligned with 14-layer EncDec, containing about 412M parameters (excluding embedding and softmax layers). During training, we perform oversampling with a temperature of 5. We list average BLEU for High, Med, Low and All language groups. We also show average BLEU and translation language accuracy (ACC) for zero-shot test sets. *D*: LMs + Deep; *W*: LMs + Wide.

set. Figure 7 shows the results. Scaling improves knowledge transfer for almost all models, while PrefixLM performs surprisingly well on zero-shot directions. In most settings, PrefixLM surpasses EncDec significantly with respect to BLEU, and such superiority is more obvious on out-of-domain evaluation and for distant language pairs.

Nevertheless, we find that PrefixLM usually underperforms EncDec in terms of PPL. In other words, EncDec still possesses the best fitting ability on zero-shot language pairs. Results on translation language accuracy explains this mismatch: compared to EncDec, PrefixLM drastically reduces off-target translation – a bottleneck of zero-shot translation (Zhang et al., 2020). This also suggests that EncDec suffers from more serious searching errors during inference (Stahlberg & Byrne, 2019), which the inductive biases of PrefixLM help.

In addition, we observe no benefits from CausalLM on zero-shot translation, with or without the source-side language modeling objective. This finding disagrees with that of Wang et al. (2021), which we ascribe to various differences in model, data and optimization. Note that Wang et al. (2021) adopted more aggressive data oversampling, didn't consider distant languages, proposed dedicated optimization with the source-side loss, used a different way to count model parameters, and designed different language tags for multilingual translation that could greatly affect zero-shot results (Wu et al., 2021). We leave the study of these differences to the future.

**How do LMs perform on massively multilingual translation?** We further examine the scalability of LMs with respect to the number of languages, and experiment on massively multilingual translation using OPUS-100. We enlarge the BPE size to 64K to handle multilingual lexicons. Following Zhang et al. (2020), we divide the test language pairs into high-resource (High, >0.9M), low-resource (Low, <0.1M), and medium-resource (Med, others) groups, and report average scores for each group. Table 3 summarizes the results. EncDec outperforms LMs on supervised directions, with larger gap on low-resource languages and for XX→En translation. By contrast, LMs, particularly PrefixLM, perform better on zero-shot directions, with improved translation language accuracy. Overall, PrefixLM outperforms CausalLM, and also performs comparably to EncDec on supervised directions (often $< -1$ BLEU on average), echoing with our above findings.

## 7 CONCLUSION AND FUTURE WORK

In this paper, we revisited language model architectures for machine translation in the perspective of model scaling and cross-lingual transfer. Extensive experiments show that LMs often have different scaling properties where the impact of architectural differences gradually reduces as model scales up, and that LMs often deliver better zero-shot transfer than its EncDec counterpart with improved off-target translation although its cross-lingual transfer on supervised directions is suboptimal. PrefixLM, the one with full visibility to the source input, shows consistent superiority to its CausalLM counterpart, and performs similarly well to EncDec across different settings especially

when paired with deep modeling. These findings show that while current product offerings for major language pairs or small on-device models should continue using EncDec, LMs can be an effective architecture for giant multilingual models with zero-shot transfer as a primary focus.

We notice that the performance gap caused by architectural differences gradually disappears as the model size increases. This has several implications for future researches: 1) Comparing NMT models in one model setting is not enough, particularly with the widely adopted 6-layer Transformer base setting, because of the scaling property difference. We believe the best practice should portray the whole scaling picture for model comparison. 2) Just like NMT models optimized for high-resource translation transfer poorly to low-resource scenarios, many models developed in the past with claims outperforming Transformer might not transfer to large-scale model settings. These studies ideally should be revisited in the face of model scaling. 3) When models are compared in a single setting due to shortage of resources, best practice should state clearly which model size the conclusion is based on. 4) PrefixLM matches the performance of EncDec at large scale, and delivers promising zero-shot transfer, which deserves more efforts to study. This is also an encouraging signal for model unification, i.e. supporting disparate tasks with a large enough yet single model (including the translation tasks), which is an exciting future direction.

We also notice that LMs underperform EncDec significantly on computational efficiency. In the future, we will put more efforts on developing more efficient networks for LM scaling. We are also interested in exploring the applicability of translation-optimized LMs to downstream monolingual and cross-lingual tasks.

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

# A    MODEL TRAINING AND INFERENCE

We update model parameters via Adafactor (Shazeer & Stern, 2018) with label smoothing of value 0.1, and scheduled learning rate of warmup steps 40K. We apply dropout of 0.1 to residuals, feed-forward activations and attentions. We employ the post-norm Transformer by default; for some exceptional cases (often with deep models where training is unstable) we use the pre-norm one instead. Batch size is set to about 128K tokens. We train models for up to 1M steps on different tasks, except Web En-De where 500K steps is used. We average 10 checkpoints for evaluation. For bilingual experiments, these checkpoints are selected according to the dev set performance; for multilingual experiments, we use the last 10 checkpoints. Beam search is used for inference, with a beam size of 8 and length penalty of 0.5.

# B    MORE EXPERIMENTAL RESULTS

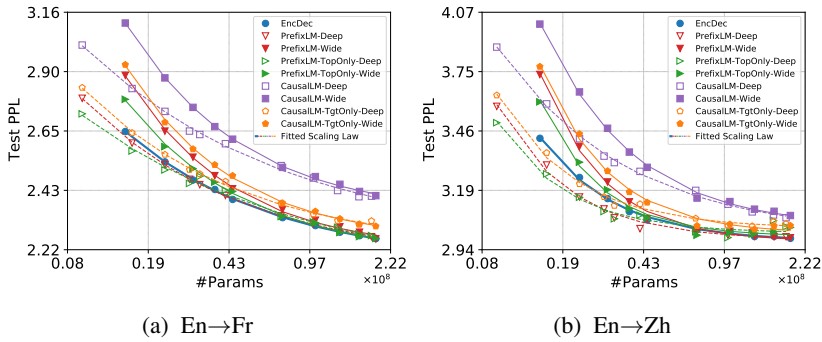

(a) En→Fr               (b) En→Zh

Figure 8: Fitted scaling curves for different models on WMT14 En-Fr and WMT19 En-Zh on **the longest sentence group**. We rank our test set according to source sentence length, and then split it into 8 disjoint groups. This shows the results on the longest group.

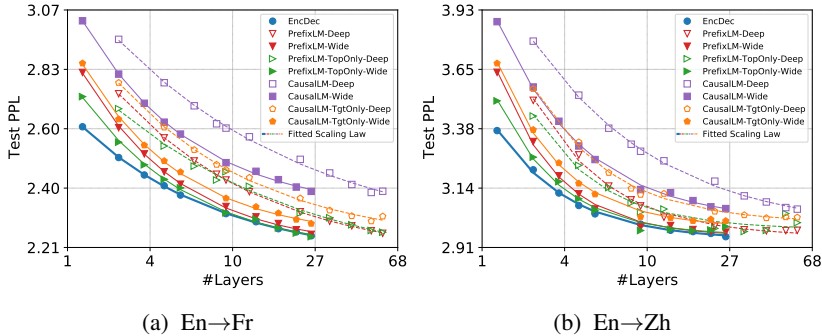

(a) En→Fr               (b) En→Zh

Figure 9: Fitted scaling curves for different models on WMT14 En-Fr and WMT19 En-Zh with respect to **the number of layers**. Note under the same number of layers, *LM + Deep* has much fewer parameters than EncDec and *LM + Wide*. The performance gap also narrows as model scales up.

# C    RELATIVE VS. ABSOLUTE TRANSFER RESULTS

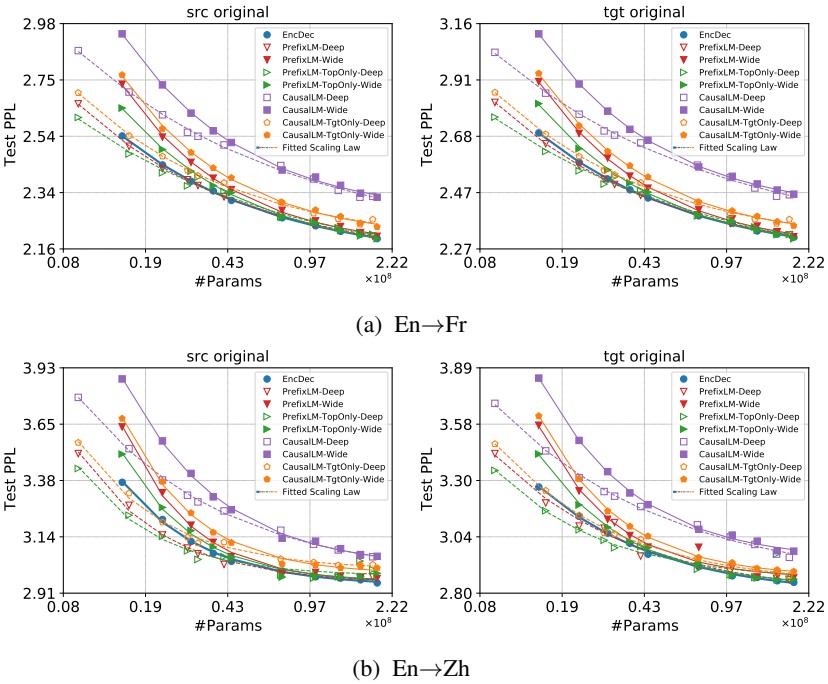

(a) En→Fr

(b) En→Zh

Figure 10: Fitted scaling curves for different models on WMT14 En-Fr and WMT19 En-Zh evaluated on *source original* and *target original* test sets.

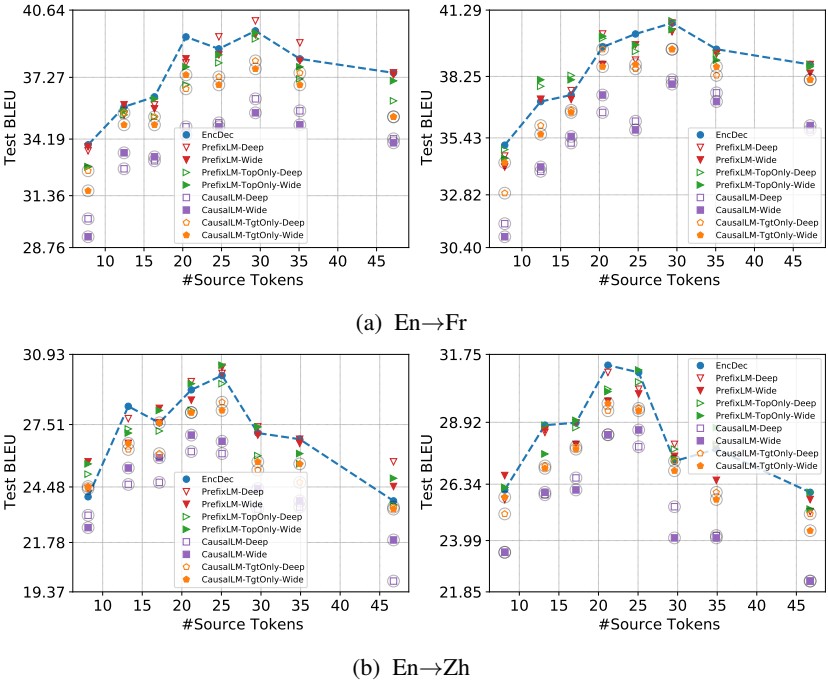

(a) En→Fr

(b) En→Zh

Figure 11: BLEU scores for different models on WMT14 En-Fr and WMT19 En-Zh as a function of source sentence length. *Left*: models aligned with 6-layer EncDec; *Right*: models aligned with 14-layer EncDec.

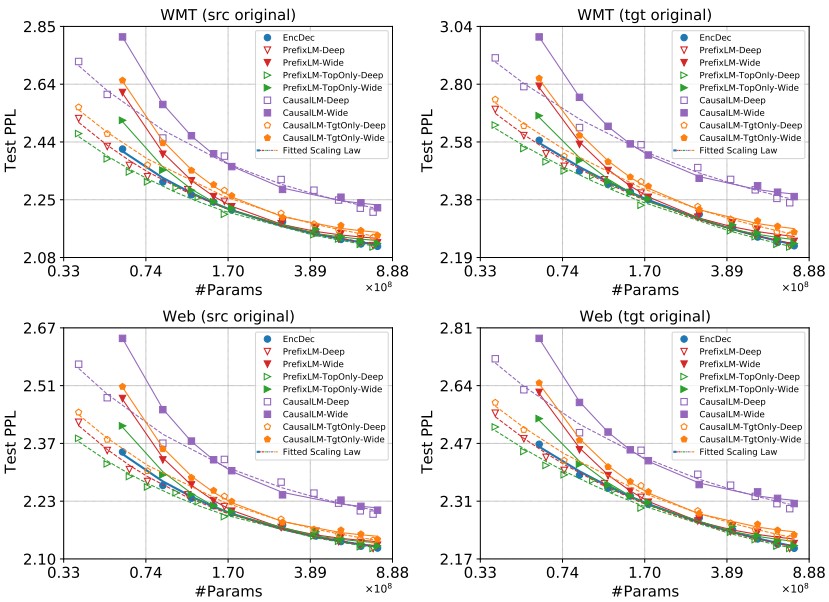

Figure 12: Fitted scaling curves for different models on Web En-De (En→De). *src/tgt*: source/target; *WMT*: out-of-domain evaluation set; *Web*: in-domain evaluation set. Models are trained in the Transformer big setting.

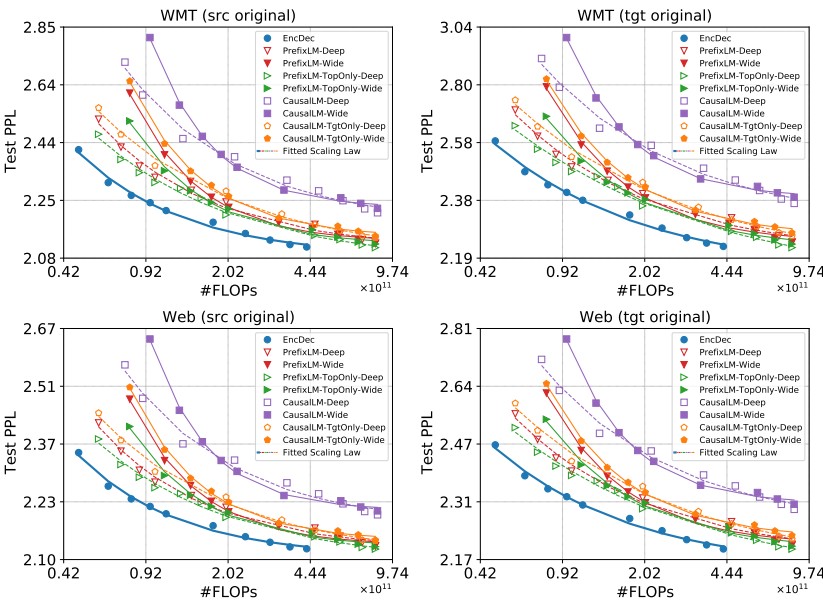

Figure 13: Fitted scaling curves for different models on Web En-De (En→De) in terms of FLOPs. Models are trained in the Transformer big setting.

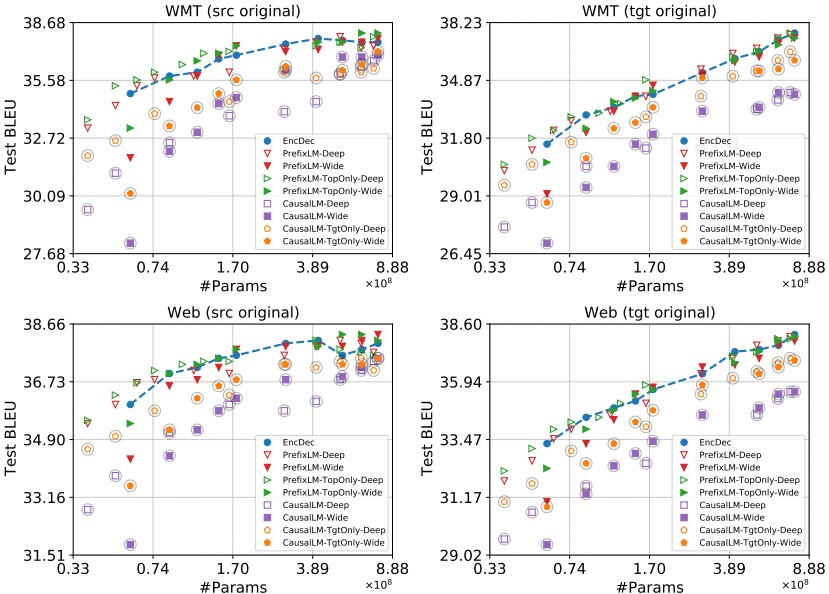

Figure 14: BLEU scores for different models on Web En-De (En→De) as a function of model parameters. Models are trained in the Transformer big setting.

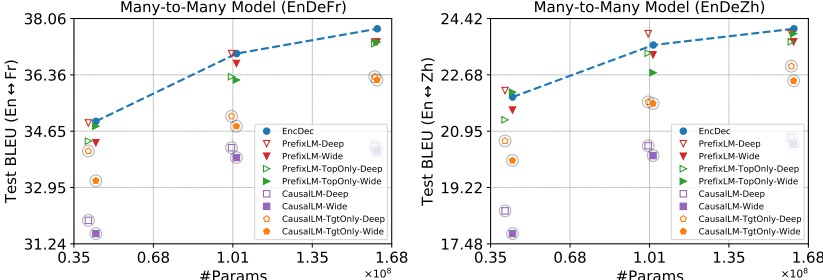

Figure 15: Cross-lingual transfer results (average BLEU scores) for different models from the low-resource language (En-De) to high-resource directions under different model sizes on WMT datasets. Average is performed over En↔Fr/Zh. **Left**: multilingual En-De-Fr system; **Right**: multilingual En-De-Zh system.

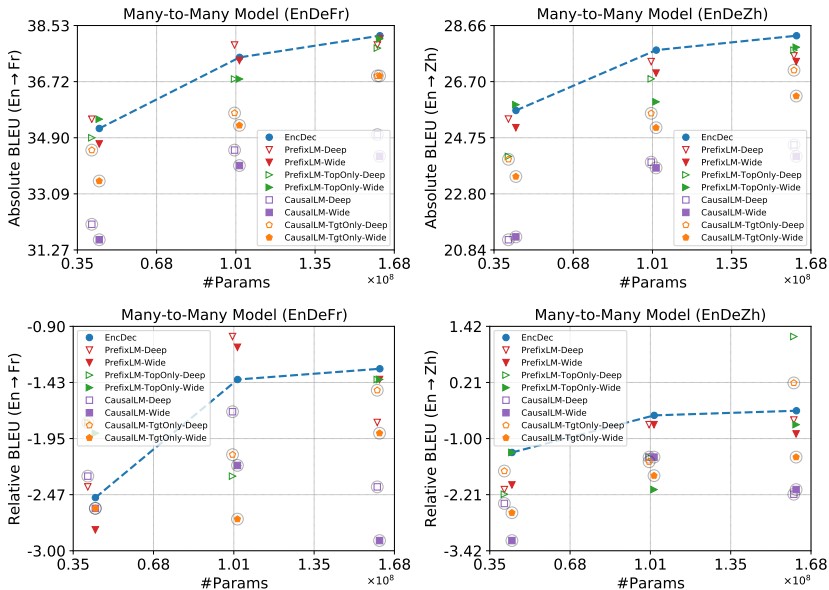

Figure 16: Absolute (top) and relative (bottom) transfer results of different models for En→Fr and En→Zh under different models sizes on WMT datasets. **Left**: multilingual En-De-Fr system; **Right**: multilingual En-De-Zh system. Relative score is computed by comparing multilingual model and its corresponding bilingual counterpart. Overall, there is no clear pattern supporting that LMs encourage knowledge transfer better than EncDec.

