# OpenReview forum: "Examining Scaling and Transfer of Language Model Architectures for Machine Translation"
_ICLR.cc/2022/Conference — ICLR 2022 Submitted_

### Official Review · Reviewer_vGrd · 2021-10-29

**Correctness:** 3
**Technical Novelty And Significance:** 3
**Empirical Novelty And Significance:** 3
**Recommendation:** 6
**Confidence:** 4

**Main Review:**

Strengths:
The authors delivered these findings via extensive experiments on different language pairs, data scales, model sizes, and translation task settings (bilingual, multilingual, zero-shot, etc.), so these findings have good reliability and generalization. Some of the findings are inspiring to the follow-up research on this topic.

Weaknesses:
Some experiments require more careful design to make findings more reliable. Such as, in Figure 2(d), the Test Bleu of EndDec, CasualLM-Deep, and Prefix-TopOnly-Deep decrease as the parameters increase in the interval of #Params 0.19-0.43E8. This phenomenon is not in line with the general trend, also the intuition. I suspect that the experimental settings could be improved. Othervise, the authors need to give a reasonable explanation.

There is another question I would like to promote a discussion. That is how to measure the transferring capabilities. The conclusions drawn from Figure 6 is that LMs do not really facilitate the transfer to low-resource languages, because LM-based models do not outperform EncDec model in all #params scales. However, we cannot see the improvement each model gain through transferring over the baseline. These relative improvements over the baseline (not the absolute scores) may be more appropriate to measure the transferring capabilities.

**Summary Of The Paper:**

There are more and more work using language model to conduct translation between languages. This paper examines scaling and transferring laws when implementing translation with LM architecture through extensive experiments. Some of these findings are straightforward and intuitive but not experimentally verified before, and some are not so obvious. The latter is especially important for the community. These findings include: LMs are generally inferior to the EncDec architecture in terms of MT performance; it does not have the advantage of speed either; while on the zero-shot transferring aspect, LMs often performs better than EncDec models; different LM architectures have different scaling properties, among which architecture differences usually have a significant impact on the performance of small-scale models, but the performance gap narrows as the number of parameters increases, and so on. We can find more in the paper.


**Summary Of The Review:**

This paper has conducted extensive experiments to examine the scaling and transferring laws of LMs for machine translation and has concluded several interesting findings which could be inspiring to the future work.

---

> ### Author Response · Authors · 2021-11-19
> **Response to Reviewer vGrd**
>
> Thanks for your insightful comments. We address each of your concerns as below.
>
> * **Regarding the BLEU Reduction in Figure 2(d)**
>
>   Thanks for pointing this out! From our understanding, such reduction indicates a mismatch between test log-perplexity and BLEU scores, which mostly comes from the originality of test sentences (particularly source-original sentences).
>
>   Please notice that Figure 2(d) shows the BLEU scores on the newstest2019 En-Zh test set, and this test set is source-original alone (this is an intended setting by WMT organizers). We also include their corresponding log-perplexity scores in Figure 2(b), where the log-perplexity decreases relatively smoothly. The decrease of test log-perplexity indicates the improvement of model fitting. However, recent studies, such as [1], suggest that the relationship between BLEU (also BLEURT) and log-perplexity is non-trivial on source-original test sets, where higher log-perplexity sometimes corresponds to higher BLEU scores. Our results further confirms such abnormal behavior.
>
> * **Regarding How to Measure the Transferring Capability**
>
>   This is a nice suggestion! We agree that reporting relative performance against bilingual baselines would be a better metric to reflect transferring capability. We can’t show the results on low-resource settings (En-De) immediately since it requires about 54 experiments on WMT En-De datasets. We hope that we can finish it soon.
>
>   However, we do have the results for high-resource settings (En->Fr and En->Zh), which we have added in the end of our updated version. There are some differences compared to absolute scores but we observe no convincing patterns which support that LMs encourages knowledge transfer more.
>
>
> [1] Scaling Laws for Neural Machine Translation. Ghorbani et al., 2021

---

> > ### Author Response · Authors · 2021-11-27
> > **Any further questions?**
> >
> > Thanks for your insightful comments again! Please feel free to add any follow-up questions!

---

### Official Review · Reviewer_AF5b · 2021-11-02

**Correctness:** 3
**Technical Novelty And Significance:** 2
**Empirical Novelty And Significance:** 3
**Recommendation:** 5
**Confidence:** 4

**Main Review:**

**Strengths**

1. The empirical evaluations of this paper are comprehensive and solid, the results and findings are interesting.
2. The paper is well-organized and clearly written.


**Weaknesses**

1. Despite good empirical efforts, the motivation of this paper seems somewhat unclear. Given that the encoder-decoder paradigm dominantly governs machine translation (also from the experiment part of this paper that LMs underperform EncDec most of the time.), for what reason should we need to consider a shift to a unified language model for such a seq2seq task? If the zero-shot transfer is the case, why not directly fix the off-target issue for EncDec and preserve the good of translation performance, instead of changing the paradigm? I feel like the authors didn't convey an incentive for this.
2. The paper conducted extensive experiments to show how scaling affects LMs for MT, however, few suggestions based on the findings are given for future development of MT.
3. The setting of CausalLM seems a bit weird that a unidirectional encoding behavior makes obviously no sense.

**Questions**
1. Section 5 seems to mainly examine p and L_{\inf} In equation (7), whereas \alpha remains undetermined. How was the value of \alpha determined for diagrams in Figure 3? Get fitted from Figure 2 I guess?


**Summary Of The Paper:**

This paper empirically investigates the scaling and transfer of Transformer language model architectures (i.e., a unified LM architecture rather than encoder-decoder models) for machine translation. The paper examines choices of architectural designs wrt data scales and model sizes and evaluates by performance (BLEU) on bilingual, multilingual and zero-shot translation, as well as the power-law (Kaplan+20) of neural language models.  Experiments and analyses are extensive and findings are interesting.



**Summary Of The Review:**

This paper is a good empirical work investigating the unified LM for machine translation and shows some good findings. However, the motivation yet remains unclear, plus few insights are delivered. I feel like the paper is not bad in terms of empirical study, but feel a bit not interesting regarding the bar of ICLR. I tend to consider it as a borderline paper but are willing to change my mind if I did miss something upon the author response.

---

> ### Author Response · Authors · 2021-11-19
> **Response to Reviewer AF5b (Part 1/2)**
>
> Thanks for your insightful comments. We will make our motivation on unification and suggestions to future development of MT more clear in our revised version. Below we address each of your concerns.
>
> * **Regarding Our Motivation on Model Unification**
>
>   **We’d like to reemphasize why we explore model unification for NMT here in two aspects.**
>
>   - Firstly, recent studies on large-scale pretraining and model scaling show the promise of moving to unified neural architectures and that dropping various task-specific inductive biases improves model generalization and/or performance [1,2]. In neural machine translation, one essential task-specific inductive bias is to separately handle source sentence understanding and target sentence generation with the encoder-decoder paradigm. Although such bias significantly benefits translation, some recent work shows insights challenging it, for example, aggressively simplifying the decoder yields little to no compromise on translation quality [3]. This thereby inspires the research question how the removal of this separation, i.e. using single unified LM modules, works for translation, and whether we can get any benefits out of that.
>
>   - Secondly, unified architecture shares parameters across tasks (languages in our setup), which brings in a distinct (dis)advantage to handle knowledge transfer. It’s surprising for us to observe that PrefixLM largely mitigates the off-target translation problem [4] and yields consistent zero-short performance improvement, and that such benefit is domain-robust. In addition, unified LMs jointly model sentence understanding and translation, which has high potential to deliver large impact to wide downstream NLP applications, and also absorb knowledge from other related NLP tasks to strengthen MT. This is particularly significant along with model scaling (as shown in our paper), and also forms our follow-up questions to be answered.
>
> * **Regarding Zero-Shot Translation**
>
>   Although recent studies have made progress on improving zero-shot translation, most of them rely on authentic or pseudo parallel training corpus, and our understanding on why the off-target translation issue occurs and how to solve it through better architecture and optimization is still limited. Compared to developing dedicated methods to improve zero-shot translation, from our perspective, figuring out which sort of architectural inductive biases favor zero-shot transfer is equally (or even more) interesting and important. And our results show that PrefixLM encodes such kinds of inductive biases, which should give the community more insights on the zero-shot transfer.
>
> * **Regarding Suggestions to Future Development of MT**
>
>   **There are many direct and indirect implications that can be drawn from our findings for future MT.**
>
>   - Firstly, model scaling matters a lot when comparing LMs and EncDec, which suggests that comparing different models in a single model configuration could be very misleading, and that future MT studies should show the whole scaling picture to reach meaningful/convincing conclusions. This is very critical for studies developing new model architectures and aiming for state-of-the-art performance, since it’s highly likely that the performance gains yielded in small settings disappear after scaling models up.
>
>   - Secondly, there are multiple findings that can be used directly as guidance for future NMT development. For example, with respect to machine translation, allowing full visibility to the source input works better than unidirectional source encoding; adding more non-linear operations via increasing model depth often benefit translation more than increasing model width; also, incorporating source-side language modeling objectives in CausalLM helps translation little. A big picture is that the performance difference of different LM settings compared to EncDec narrows as the model scales up.
>
>   - Lastly, PrefixLM greatly enables the reduction of off-target translation and improves zero-shot performance. It also yields comparable performance to EncDec on supervised directions. We believe that PrefixLM deserves more efforts for future NMT. In addition, as we mentioned in our conclusion, our findings also suggest that while current product offerings for major language pairs or small on-device models should continue using EncDec, LMs can be an effective architecture for giant multilingual models with zero-shot transfer as a primary focus.

---

> > ### Author Response · Authors · 2021-11-19
> > **Response to Reviewer AF5b (Part 2/2)**
> >
> >
> > * **Regarding The Setting of CausalLM**
> >
> >   CausalLM follows the traditional formulation of language modeling, which is an important member of the language model family. We add the comparison with CausalLM such that we have a complete picture for different LMs against EncDec, which also makes our study convincing.
> >
> > * **Regarding $\alpha$ in Equation 7**
> >
> >   As explained under Equation 7, $\alpha$, $p$ and $L_{\inf}$ are all fitted parameters based on the scaling results. We will release our source code to show how we compute them.
> >
> >
> > [1] Language Models are Few-shot Learners. Brown et al., NeurIPS 2020.
> >
> > [2] Scaling Laws for Neural Language Models. Kaplan et al., 2020.
> >
> > [3] Deep Encoder, Shallow Decoder: Reevaluating Non-autoregressive Machine Translation. Kasai et al., ICLR 2021.
> >
> > [4] Improving Massively Multilingual Neural Machine Translation and Zero-Shot Translation. Zhang et al., ACL 2020

---

> > > ### Author Response · Authors · 2021-11-27
> > > **Any further questions?**
> > >
> > > Thanks for your insightful comments again! Please feel free to add any follow-up questions!

---

### Official Review · Reviewer_tsgN · 2021-11-02

**Correctness:** 4
**Technical Novelty And Significance:** 2
**Empirical Novelty And Significance:** 2
**Recommendation:** 3
**Confidence:** 5

**Main Review:**

This is an OK paper. The writing and presentation are clear, the experiments are relatively thorough, the related works are discussed comprehensively, the results are convincing, the claims are well supported by the results.

However, after a careful reading of the paper, I still could hardly find enough insights from the paper. ***How could the findings benefit the community?*** The design principle of EncDec-based NMT is the sub-optimal performance/efficiency of LM-based NMT, and this paper just says: "yes, the intuition is true", which is not exciting at all. One insight is the zero-shot performance of multilingual NMT, however, I do not think there are many differences between 4.80 BLEU scores and 7.95 BLEU scores, where are too low to make sense.

It would be nice if the authors could discuss the applicability of the findings in the author response and the future version. And better yet, directly utilizing the findings to enhance existing NMT systems. Furthermore, the paper topic is too narrow, it would be much better to extend to other language generations tasks, like dialogue and QA (not a weakness but a suggestion).



**Summary Of The Paper:**

This paper gives a quite thorough comparison between language-model-based (LM) and encoder-decoder-based (EncDec) architectures for neural machine translation (NMT). The investigated LMs are varied: different masking strategies, different origins of source features, and so on. The authors conducted experiments on representative bilingual and multilingual benchmarks, summarizing some findings based on the experimental results.

**Summary Of The Review:**

This is an OK paper but the current version is not exciting, which could hardly attract attention from the community. The authors should answer the question: how could the paper findings benefit the community?

---

> ### Author Response · Authors · 2021-11-19
> **Response to Reviewer tsgN (Part 1/2)**
>
> Thank you for the insightful comments, particularly the suggestion on extending our study to other generation tasks which we will put more effort on in the future.
>
> Before elaborating our response, we hope the reviewer could clarify this comment *“The design principle of EncDec-based NMT is the sub-optimal performance/efficiency of LM-based NMT, and this paper just says: "yes, the intuition is true", which is not exciting at all.”* which is difficult for us to understand.
> * If you mean that EncDec was supposed to be worse than LMs, our results show that LMs underperform EncDec in many translation settings and often deliver poorer efficiency on FLOPs.
> * By contrast, if you mean that EncDec was supposed to be better than LMs, our results show that such superiority gradually disappears as the model scales up. Scaling matters more while architecture matters less. In addition, LMs largely outperform EncDec in zero-shot translation, and perform significantly better in handling off-target translations as demonstrated in WMT and OPUS experiments.
> In either case, we have exciting and confident insights for the readers.
>
>
> **We’d like to emphasize that our study includes insightful findings that should be interesting for wide communities.**
>
> - First of all, the comparison of EncDec and different LMs on NMT is intriguing yet missing. Large-scale pretrained models, such as LM-based BERT and EncDec-based T5 [1], have largely improved and obtained competitive performance across diverse downstream tasks. However, how EncDec and different LMs perform in NMT under different setups has not yet been adequately explored. What sorts of inductive biases benefit translation and what hurt translation are fundamental questions to be addressed, which is definitely an interesting/important topic to the community. We offered such a study in this paper to answer these questions. Besides, such comparison might also shed light on why pretraining helps high-resource machine translation little as reported in [1].
>
> - Secondly, one unexpected observation is that different models show different scaling properties, and that the performance gap caused by architectural differences gradually disappears as the model size increases. This has several implications for future studies: 1) Comparing NMT models in one model setting is not enough, particularly with the widely adopted 6-layer Transformer base setting, because of the scaling property difference. We believe the best practice should portray the whole scaling picture for model comparison. 2) Just like NMT models optimized for high-resource translation transfer poorly to low-resource scenarios, many models developed in the past with claims outperforming Transformer might not transfer to large-scale model settings. These studies ideally should be revisited in the face of model scaling. 3) When models are compared in a single setting due to shortage of resources, best practice should state clearly which model size the conclusion is based on. 4) PrefixLM matches the performance of EncDec at large scale. This is an encouraging signal for model unification, i.e. supporting disparate tasks with a large enough yet single model (including the translation tasks), which is an exciting future direction.
>
> - Thirdly, our study compares a broad set of inductive biases, both in the architecture design and loss being induced, that are beneficial to translation. Through our comparison, we observe that using the full-visible mask over the source input, feeding the final-layer source encoding to the decoder, and deepening models and removing source-side objective are all helpful to translation when using LMs for translation and especially when the model size is of small-to-medium scale. These findings should be interesting for researchers focusing on developing new architectures, loss functions, regularizers or optimization methods for NMT.

---

> > ### Author Response · Authors · 2021-11-19
> > **Response to Reviewer tsgN (Part 2/2)**
> >
> >
> > - Fourthly, PrefixLM substantially reduces off-target translations and improves zero-shot performance. The off-target issue is one of the main bottlenecks for zero-shot translation, but why it happens and how to handle it without accessing (authentic or pseudo) training corpus on zero-shot directions remains difficult and open questions. PrefixLM largely avoids this problem, and delivers encouraging performance under different settings (WMT and OPUS). Its success should give researchers more inspirations on architectural inductive biases that optimize the zero-shot transfer. The reviewer mentioned that the quality improvement, from 4.80 to 7.95 BLEU, makes little sense, which unfortunately we can’t agree with. The overall low translation quality here demonstrates the great challenge and difficulty of zero-shot translation in OPUS-100, where hundreds of languages are involved. The gain (+3 BLEU) is promising, particularly considering that models trained with pseudo parallel corpus just yield a BLEU score of ~11 in the same setting as shown in [2]. Apart from OPUS-100, results on WMT in Figure 7 also show great improvement by PrefixLM (+2 BLEU), particularly on out-of-domain evaluation sets. Please also notice the large improvement on the translation language accuracy (about 15%+ ACC), a strong indicator of the reduction in off-target translations.
> >
> > - Last point we can make here is that our study aims to improve our understanding by investigating the differences in the inductive biases of EncDec and different LMs in translation, rather than delivering new state-of-the-art results, in other words it is closer to basic research than applied. Our research question is clear, experimental setup is adequate and results show evidence to support our findings to the questions. We believe our findings are interesting, and could inspire different researchers in different ways.
> >
> >
> > [1] Exploring the Limits of Transfer Learning with a Unified Text-to-Text Transformer. Raffel et al., JMLR 2020
> >
> > [2] Improving Massively Multilingual Neural Machine Translation and Zero-Shot Translation. Zhang et al., ACL 2020

---

> > > ### Author Response · Authors · 2021-11-27
> > > **Any further questions?**
> > >
> > > Thanks for your insightful comments again! Please feel free to add any follow-up questions!

---

### Official Review · Reviewer_aP6J · 2021-11-02

**Correctness:** 3
**Technical Novelty And Significance:** 2
**Empirical Novelty And Significance:** 3
**Recommendation:** 6
**Confidence:** 4

**Main Review:**

This is a paper on investigating the gaps of information in a relatively new area of research in machine translation. The approach, experiment setup, and evaluation methods are reasonable and provide confident insight into the questions that the authors ask.

Here are a number of questions for the authors:

- Your TopOnly setting is supposed to mimic how the encoder-decoder model uses the topmost-layer source encodings for translation. This setting indeed shows improvements and achieves lower perplexity on almost all language pairs. One can interpret that two fundamentally different architectures are still bound by similar features. Did you investigate why the topmost-layer is the most informative way (at least in the tried approaches) of transferring information from the source to the target generation? Following on that, looks like that the difference between the _Deep_ and _TopOnlyDeep_ variations is very small. What is your understanding on the difference of information captured by deeper _and_ multiple layers versus deeper and topmost layer? Did you compare _Deep_ and _TopOnly_ setups? This question is mostly about the PrefixLM model.

-  You mention that the _TgtOnly_ setting is supposed to enrich the CasualLM model with additional information. However it
requires and occupies part of modeling capacity. Can you explain what you mean by occupying the modeling capacity?

- You conclude in one of the sections that "When there are fewer parameters, the model with inductive biases favoring translation will achieve better quality" and follow it with four observations two of which I want to discuss here. It is not entirely clear to me why "deeper LMs (_Deep_) rather than wider" and "training LMs without source-side language modeling loss (_TgtOnly_)" are both considered as high inductive biases. For the former, how is deeper models have more inductive biases than wider models? For the later, wouldn't the other way around be correct?

- You observe that the PrefixLM model performs surprisingly well on zero-shot directions. Do you have any intuitions or have you looked into finding _why_ that's the case?

- According to your experiments, it looks like CausalLM (unidirectional) model is not performing well, or at least as good as PrefixLM or EncoderDecoder models, in almost any settings. This is intuitive because the model has access to less information at each step. I'm wondering if you _can_ think of a scenario where such a model will be more effective.

- Since this is an examination work on multiple translation models and configurations, it would have been nice to provide a deeper analysis on the linguistic aspects of the experiments as well: Language pairs with different characteristics (morphologically rich languages for instance) work differently with different models. How different variations of the same model perform on different classes of languages is a significant question to investigate.





**Summary Of The Paper:**

This paper provides an in-depth examination of language model based translation models and the impact of various parameters in the learning process.

They pick the "classic" encoder-decoder based translation model as the baseline and propose several variations of language model based translation models to compare. This includes the two main categories of unidirectional and bidirectional language modelling of the source language. On top of that, they look into various setups such as which layers to use to pass the source encodings to the target generation phase of the translation, whether to share parameters between source and target, and changing the depth (number of transformer layers) and the width (feed forward dimension).

They evaluate on in-house and publicly available datasets and investigate how different setups compare in the supervised translation task, zero-shot learning, low-resource vs. high-resource language pairs, and bilingual to multilingual translation.


**Summary Of The Review:**

I find this paper an interesting read. The comprehensive study into different variations sheds light on effectiveness of language model based translation models in different settings. The approach, experiment setup, and evaluation methods are reasonable and provide confident insight into the questions that the authors ask.
This is valuable in the progress of research in developing new translation models.
One downside of this work is that there is no novel approach proposed in this paper, as well as a more comprehensive study into existing models and combining various setups is more useful to the research community.

---

> ### Author Response · Authors · 2021-11-19
> **Response to Reviewer aP6J**
>
> Thanks for your insightful comments. Our response to each of your concerns is given below.
>
> * **Regarding TopOnly Structure**
>
>   Factors that make the TopOnly variant favorable to translation tasks are many. Based on the literature [1], representations in Transformer often evolve from the bottom up, where lower-layer encodings align better with syntactic-related information while the higher-layer ones correlate more with semantic-related information. From our understanding, translation is a task that requires source-side semantic knowledge to provide clues for accurate source-target alignment, thus the top-most source encodings are preferred. This further explains the narrowed performance gap between Deep and TopOnly-Deep, since deeper layers could offer more abstract and semantic-intensive representations to the decoder to ensure the translation accuracy.
>
>   As for “Deep” and “TopOnly”, we assume that you mean Deep and TopOnly-Wide. This comparison is already included in all figures.
>
> * **Regarding TgtOnly Setting**
>
>   We believe there is some misunderstanding about the “TgtOnly” setup. By default, CausalLM uses both Src and Tgt objectives for training. TgtOnly means that we only train CausalLM with the Tgt objective. We argue that the Src language modeling objective “enriches CausalLM with additional information and occupies part of modeling capacity”. This is because by training with source-side objective, the model learns extra information about the unconditional source language structure and also its distinct linguistic characteristics. However, along with the learning, the model inevitably splits part of its capacity to support it.
>
> * **Regarding Inductive Biases of Deeper LMs and TgtOnly**
>
>   Perhaps there is another misunderstanding. The two observations pointed out here are based on our empirical results where Deep LMs outperform its Wider counterparts, and CausalLM+TgtOnly outperforms CausalLM when the model is of small scale. We conjecture that models yielding better performance often encode **better inductive biases towards translation**, rather than having more inductive biases. In other words, we treat deep modeling and wide modeling as two different types of inductive bias.
>
>     As for the TgtOnly setup, this variant includes the target-side objective alone, i.e. without source-side language modeling loss. Please refer to our above response regarding TgtOnly.
>
> * **Regarding Why PrefixLM Works Well on Zero-Shot Pairs**
>
>   Zero-shot translation is often sensitive to many factors, such as data sampling, language tagging, optimization, model architecture and so on. We didn’t explore all potential factors, which is beyond the scope of this paper. But our current study suggests that the improved performance mostly comes from the reduction of off-target translations as shown in the bottom row, Figure 7, i.e. the structure of PrefixLM could encourage the model to translate into the correct target language.
>
> * **Regarding CausalLM**
>
>   In the context of machine translation, there are certain scenarios where full source input is unavailable, such as simultaneous machine translation. Unidirectional modeling in this case is more favorable due to its efficiency (without the need to re-encode every partial source segment) and likely to be more effective because of the training-testing consistency.
>
> * **Regarding Deeper Analysis on Different Classes of Languages**
>
>   We fully agree with you: different languages have different characteristics which work differently with different models (and under different data conditions). That’s exactly why we include diverse languages for experiments, such as English, French, German and Chinese. We hope the reviewer could agree that incorporating all potential languages of interests into our study is far beyond our compute capacity and research scope.
>
> * **Regarding “more comprehensive study and combining various setups”**
>
>   Thanks for your comments! Could you please give us some concrete suggestions here? For example, which sort of studies and setups do you believe would benefit our work?
>
>
> [1] BERT Rediscovers the Classical NLP Pipeline. Ian et al., ACL2019.

---

> > ### Author Response · Authors · 2021-11-27
> > **Any further questions?**
> >
> > Thanks for your insightful comments again! Please feel free to add any follow-up questions!

---

### Decision · Program_Chairs · 2022-01-20

**Decision:**

Reject

**Comment:**

This paper has conducted extensive experiments to examine the scaling and transferring laws of LMs for machine translation and has concluded several interesting findings which could be inspiring to the future work.

The main concerns from reviewers are that the novelty of this paper is not enough. In addition, the experiments are not well-designed and the clarity of this paper can be further improved. We hope the reviews can help authors improve their paper.